# Welfare Assessment Tools in Zoos: From Theory to Practice

**DOI:** 10.3390/vetsci9040170

**Published:** 2022-04-01

**Authors:** Narelle Jones, Sally L. Sherwen, Rachel Robbins, David J. McLelland, Alexandra L. Whittaker

**Affiliations:** 1School of Animal & Veterinary Sciences, The University of Adelaide, Adelaide, SA 5371, Australia; dmclelland@zoossa.com.au (D.J.M.); alexandra.whittaker@adelaide.edu.au (A.L.W.); 2Wildlife Conservation and Science, Zoos Victoria, Melbourne, VIC 3052, Australia; ssherwen@zoo.org.au; 3The Animal Welfare Science Centre, The University of Melbourne, Melbourne, VIC 3052, Australia; 4Zoos South Australia, Adelaide, SA 5000, Australia; rrobbins@zoossa.com.au

**Keywords:** zoo animal welfare, five domains, validity, animal-based, resource-based, scoring

## Abstract

Zoos are increasingly implementing formalized animal welfare assessment programs to allow monitoring of welfare over time, as well as to aid in resource prioritization. These programs tend to rely on assessment tools that incorporate resource-based and observational animal-focused measures. A narrative review of the literature was conducted to bring together recent studies examining welfare assessment methods in zoo animals. A summary of these methods is provided, with advantages and limitations of the approaches presented. We then highlight practical considerations with respect to implementation of these tools into practice, for example scoring schemes, weighting of criteria, and innate animal factors for consideration. It is concluded that there would be value in standardizing guidelines for development of welfare assessment tools since zoo accreditation bodies rarely prescribe these. There is also a need to develop taxon or species-specific assessment tools to complement more generic processes and more directly inform welfare management.

## 1. Introduction

In recent years the welfare of animals held in zoos has been of increasing interest to the public, with an expectation that high standards of welfare are achieved [1,2]. It has been suggested that even with a commitment to species conservation, the housing of animals in captive environments is not justified unless high standards of animal welfare are apparent [3]. Animal welfare has also become a higher priority within the zoo industry itself [4]. As such, there has been a recent impetus on development and validation of methods to assess the welfare of zoo animals. Furthermore, formal assessment of welfare status has recently received increased focus as a component of zoo accreditation schemes, such as that of the Association of Zoos & Aquariums in the US [5], and the Zoo and Aquarium Association in Australasia [6] whose program is based around the Five Domains model for welfare assessment [7]. Institutions within these associations are required to complete and maintain industry accreditation, which requires formal processes for ongoing review and enhancement of animal welfare. These accreditation processes should be evidence-based and informed by research in welfare science.

There are multiple methods used to assess animal welfare, including assessment of physiological, immunological, or behavioral responses elicited in response to an intervention or housing environment [8,9]. Whilst it is commonplace to combine methods to enable a holistic determination of welfare state, methods for assessment of zoo animals should be non-invasive and undemanding on resources. Behavior-based methods are therefore likely to be the most practical to apply. A further consideration in selecting relevant indicators is the balance between resource-based and animal-based measures. The former relate to the animals’ environment and are often referred to as ‘inputs’. They may include determinations of space allowance, temperature or humidity, or appropriate food presentation and nutritional value. They are generally quantitative, highly repeatable across different observers, and easy to record. However, these measures may not be correlated with the actual affective state or condition of an animal [3,10]. A recent shift in the trend of welfare science has seen the supplementation of resource-based assessments with animal-based indicators [10,11]. Animal-based indicators directly measure a combination of physical, behavioral, and physiological variables, and also consider the varied response of individuals to the same provision of resources [3,10]. Animal-based indicators are considered superior since they can provide more direct information on animal-affective state [12]. From a practical viewpoint, a combination of both resource- and animal-based indices is likely to provide a more holistic assessment of animal welfare, whilst being reasonably practical to implement [13].

It is now well established that consideration should be given to the promotion of positive mental states, as well as the assessment of both positive and negative states [14]. It should also be noted that the presence of negative metal states does not necessarily equate to poor welfare. For example, it has been suggested that many species may be required to overcome short-term challenges, which may incite temporary negative states, in order to experience fundamental and lasting levels of good welfare [15]. Creating the opportunity for animals to experience positive states requires that animals have agency, and can therefore make choices about their activities and have a level of control over outcomes in their life [16]. This may even lead to a positive feed-forward mechanism where animals with positive welfare are more likely to engage in activities that enhance their welfare [17,18]. Whilst this shift in focus towards more positive states is widely accepted, it is generally regarded as challenging since current assessment methods are relatively more extensive, and developed, for the assessment of negative emotions such as pain or fear than they are for positive emotions. Furthermore, there has been less research attention directed towards identification of indicators of positive emotional state in some phyla, such as reptiles or amphibians, and even some mammalian species, in comparison to other more common domesticated species.

In this narrative review, a ‘how-to’ guide to creating welfare assessment tools for zoological parks is proposed. This guidance is based on an overview of the various theoretical foundations that have been used to derive welfare assessment tools for zoo animals, together with discussion of the considerations necessary to turn these into practical and quantifiable instruments. The latter includes aspects such as scoring schemes, weighting of criteria, validity/reliability, and the split between animal- and resource-based assessments, whilst considering factors such as seasonality, innate behaviors, and individual animals’ experiences.

## 2. Review Methodology

The review methodology was not systematic in nature given the breadth of the review question with no clear intervention or outcome criteria. However, structured searches were performed in the following databases: CAB Abstracts, Web of Science, and Scopus. Keyword search terms used included: ‘zoo’ and ‘welfare assessment’, ‘captive animal’ and ‘welfare’, ’zoo’ and ‘well-being’, ‘zoo’ and ‘welfare tool*’, ‘welfare tool’ and ‘scoring’. In addition, the reference lists from sourced papers were scanned for additional citations, and forward citation searching was performed on papers identified.

## 3. Defining Welfare and Emotion

There is no definitive definition for the term ‘animal welfare’; however, it is generally considered that ‘animal welfare’ comprises multiple physical, behavioral, and psychological elements. Welfare may be defined as “how an animal is coping with the conditions in which it lives. An animal being in a good state of welfare if it is healthy, comfortable, well-nourished, safe, able to express innate behavior, and not suffering from unpleasant states such as pain, fear and distress” [19]. Welfare is generally considered to be a long-term state that is made up of the sum of experiences of an individual. These individual experiences are affective states, described in terms of their characteristics of valence and arousal. Affective states have been variously described as an animal’s feelings, emotions, or moods. They are commonly described in terms of their dimensions which comprise (1) direction of effect, i.e., positive or negative (valence) and (2) arousal (level of activation) [8]. Individuals experiencing mostly positive affective states, such as reward, are said to have good welfare. Alternatively, animals that mostly experience negative affective states such as pain or fear are described as having poor welfare [20]. As previously discussed, in spite of the availability of a wide variety of physiological and other biological methods for ascertainment of the nature of affective states [21], these are commonly invasive or impractical to implement in a zoo setting on a regular basis. Consequently, behavioral and resource-based measures predominate and are the focus of this review as a basis for performing institutional based welfare assessments of animals.

The use of behavioral indicators is a practical, non-invasive method of measuring affective states in zoo animals [11]. However, the difficulty lies in identifying and validating taxon or species-specific indicators and relating them to effect. It is generally regarded that for behavior to be a useful indicator, there is a need for animal caretakers to be familiar with the full range of normal behaviors for the species being assessed [22]. The behavior must also occur frequently enough for identification to be practical. For many species held in zoos there is limited literature on natural behavioral biology, and thus determining useful indicators can be a challenge [22]. Facial expressions, vocalizations, social interactions, and ‘opportunity situations’ such as inquisitive exploration and play are examples of behavioral indicators of good welfare [11,13] with strong potential for inclusion in zoo animal assessment tools due to their practicality and non-invasive nature. However, many of these have not been characterized in zoo species. Moreover, ideally there should be empirical evidence that the behavior relates to the affective valence of the animal, and the direction of that affect [23]. In the absence of this direct evidence, those creating assessment tools should consider how they can demonstrate the link between the behavior and effect with inductive reasoning [23].

It is important to note that whilst behavioral measures are practical and inexpensive indicators of welfare, caution should be taken when interpreting observations. Many species do not clearly express behavioral signs of poor welfare [11] and some have adapted methods to avoid displaying signs of pain or distress [24]. The latter may be particularly important since displaying signs of stress or illness may lead to a loss of social standing within a group or could result in the animal being targeted as prey. Other species may express behavioral indicators, but individuals may display varied responses to the same welfare compromise [22]. Furthermore, when behavior monitoring is incorporated into an assessment tool the behavior should be fully defined, including the context in which it may be seen. A behavior that may occur in different contexts may then legitimately be included in different functional categories [23]. Hence, where possible, a combination of behavioral and practically obtainable physiological and biological or health markers should be considered when developing a welfare assessment tool [11]. Use of established, validated assessment frameworks will assist in achieving this.

## 4. Frameworks and Protocols Utilized to Inform Welfare Assessment in Zoo Animals

The five freedoms were first proposed by Brambell 1965, and are listed by the FAWC as freedom from hunger and thirst; freedom from discomfort; freedom from pain, injury, and disease; freedom from fear and distress; and freedom to express normal behavior [25]. These freedoms have been widely used to describe welfare [13] and are still used today, particularly in legislative documents [26]. However, the limitation of this approach is that freedom from a negative state does not necessarily imply that the animal is experiencing a positive state. In recent years the development of the Five Domains model has addressed some of the deficiencies with the Five Freedoms model, paying greater consideration to positive experiences and the mental state of an animal [27]. Legislative reform efforts are also including consideration for the Five Domains in the drafting of updated statutes and delegated legislation (see, e.g., [28]).

These frameworks have become the foundation for the development of a range of welfare assessment tools across multiple species of captive animals [29], including those held in zoos. Named models are described below with a brief overview of their use in relation to animals in zoos (see Table 1 for summary).

### 4.1. Five Domains Model for Animal Welfare Assessment

The Five Domains model for animal welfare assessment references four physical domains; nutrition, environment, health, and behavior. The fifth domain, mental state, enables the assessment of the animal’s overall affective state resulting from the physical domains described [7]. In spite of the adoption of the Five Domains by the World Association of Zoos and Aquaria, and the Zoo and Aquarium Association of Australasia (ZAA) as the framework for welfare assessment and consequent accreditation under their schemes, there is little published literature on the practical implementation of this framework in zoological parks.

The largest scale adoption of this framework published in relation to zoo animals was that carried out by Sherwen and colleagues 2018 [3], where zoo-wide use was trialed across 339 species in three zoos. In this study, the authors used the Five Domains model as a tool to conduct institutional-level assessment of welfare risk factors, as opposed to individual animal welfare monitoring. The former type of assessment may be better able to inform resource allocation and have greater application in benchmarking [3]. The resulting process has value in highlighting areas of risk for suboptimal welfare, whilst simultaneously providing opportunities for improvement. However, the method may not be suitable for use as the sole process for animal welfare assessment. This is because the process derived was largely focused on assessment of resources or inputs and only included a few animal-based risk factors, and whilst provision of resources may translate to good welfare, this is not always accurate. Furthermore, the tool is designed as a risk assessment, not a direct welfare-assessment process. It is also important to note that there is still a relative lack of information on indicators of affective state in a number of animals housed in zoos, particularly reptiles and amphibians. Hence, supplementary processes or research are required to gather these data to incorporate into the Five-Domains-based tool [3].

### 4.2. The European Welfare Quality^®^ Animal Welfare Assessment Protocol

The European Welfare Quality^®^ (WQ) animal welfare assessment protocol was originally designed to assess the welfare of production animals based on four principles, similar to those in the Five Domains model. The four principles of good feeding, good housing, good health, and appropriate behavior, are defined by a set of 12 criteria deemed meaningful to the animals [30]. These criteria are allocated measurable indicators based on the opinion of expert consultants. An important aspect is the three-step approach to scoring where (1) welfare measures, for example body condition score or space allowance, make up criteria scores; (2) criteria scores integrate into scoring for the four principles; and (3) an overall assessment of the enterprise is made based by combining the principle scores [31]. However, the practicality of this protocol is significantly reduced due to the amount of time and resources required for implementation, largely as a result of the number of criteria [10]. For example, there was inherent redundancy in production animals, with welfare classification being accounted for using only two resource-based measures [32,33]. Further testing and validation of the tool may lessen this.

In their adaptation of the protocol to farmed foxes and mink, Mononen et al. (2012) provide a useful discussion of the considerations for adaptation of the protocol to other species [31]. A key point is that whilst some welfare measures may be relevant for multiple species, sets of measures and how their evaluation feeds into criteria scores, i.e., their weighting, is species-specific. Therefore, welfare indicators that are valid, feasible, and reliable for the species of interest are first required. These may be derived from literature review or expert opinion using Delphi techniques (see later). The protocol emphasizes the desirability of including as many animal-based measures, as opposed to resource-based measures, as possible [34]. Where this is not feasible there is recognition that it may be appropriate to use an input-based measure that shows a good correlation with an animal-based measure [34]. The welfare measures selected should also be independent of each other [35].

The first adaptation to animals in a zoo setting was a Cetacean Welfare Assessment process (C-Well), applied to bottlenose dolphins [36]. The measures were derived from published literature on dolphin health, behavior, and physiology based on how these measures differentiated normal from abnormal states. Advice from marine mammal specialists, veterinarians, and welfare scientists was incorporated, resulting in measures that comprised those recorded whilst animals were working with a trainer, observations when not with a trainer, opportunistically derived measures, and those obtainable from records or questioning of the curator. The tool developed was considered to be useful for assessment of individual welfare, chosen demographics, and whole facilities, and was practical to implement, although in this first iteration it was suggested that it be performed by individuals trained in the methodology and metrics. The WQ has since been applied successfully to Dorcas gazelles [37], and proposed for the pygmy blue-tongue skink [10].

### 4.3. Zoo-Specific Welfare Assessment Programs

In 2015, Kagan and colleagues proposed a universal animal welfare framework for zoos. The authors describe four main components that together aim to provide excellent animal well-being. These are: institutional philosophy and policy, programmatic structure and resources, execution, and evaluation [16]. The framework recognizes that exemplary standards of care for animals do not necessarily lead to optimum welfare. Instead, there is a need for transparency, discussion, accountability, and ongoing commitment and evaluation of policy. An important part of this is ensuring that resource allocation is adequate. Various factors play into the success of the proposed framework, including staff training, an animal welfare communication process to include all stakeholders, welfare leadership, and a tiered evaluation process with internal and external welfare evaluations. The strength of this program is the holistic approach taken with the explicit recognition of a multitude of factors that impact on animal welfare and the assessment of it in an institutional context.

An Opportunities to Thrive program was created at San Diego Zoo Global (SDZG). This program specifically addresses the complexities of conservation breeding programs which need to consider optimizing welfare whilst increasing the chances of successful reintroductions to the wild [38]. The program uses the Five Freedoms but transforms them from a focus on reducing negative indicators to the attainment of positive affect. For example, freedom from pain, injury, and disease is transformed to the “opportunity” to achieve optimal health. Desired outputs or ways of assessing the criteria are defined, and these are targeted by management inputs that would help to achieve these; for instance, the availability of health checks. In this way a structure is given to efforts to assess and improve welfare and the resulting program considers welfare outcomes in the context of management inputs. However, as with other assessment methods, deriving suitable outputs can be challenging when there is a lack of knowledge about species-typical biology or behaviors.

Many zookeepers spend a significant amount of time interacting with and observing individual animals in their care. Evidence exists that keepers can indeed detect subtle changes in an animal’s behavior and condition, and can also reach high levels of inter-rater reliability when assessing behaviors or individual animal traits that may reflect underlying welfare states [39,40,41]. Furthermore, keeper assessments have been validated by correlating science-based welfare indicators such as fecal glucocorticoid metabolite (FGM) concentration with keeper ratings of traits. For example Wielebnowski et al. (2002) found that individual clouded leopards that were rated highly by keepers on behaviors such as “tense” and “stereotypic pacing” had higher mean concentrations of FGM than individuals that received low scores for these items [42]. In an attempt to more systematically collate these keeper insights into a welfare tracking tool, The Chicago Zoological Society developed the WelfareTrak^®^ system, a tool that is designed to use keeper assessments to monitor the well-being of individual animals over time by completing brief surveys on a weekly basis [41]. The surveys include a range of 10–15 indicators designed to reflect both the physical and emotional welfare of the animals and the website tracks responses over time, highlighting any changes in welfare score.

The animal welfare assessment grid (AWAG) is another tool originally developed for monitoring the welfare of animals involved in research that has more recently been applied for use in zoo animals [43]. The AWAG process uses caretaker ratings with a focus on cumulative lifetime experience. It records physical health, procedural events, environmental comfort, and assessments of psychological wellbeing; is computer based; and generates a numerical and visual representation of animal welfare [13]. In the first use of the method in practice (as opposed to applying it retrospectively) Brouwers and Duchateau (2021) investigated its usefulness for two groups of zoo-housed western lowland gorillas over a few months by comparing data from behavioral observations with data from keeper assessments of animal-based indicators [44]. Examples of indicators within the domains included general condition and clinical assessment (physical domain); sedation and change to routine (procedural); housing and group size (environmental domain); abnormal behaviors; and enrichment use (psychological domain). They found keeper assessments were able to capture more subtle changes in welfare, but these scores did not always correspond with data from behavioral observations.

**Table 1 vetsci-09-00170-t001:** Summary of welfare assessment frameworks and protocols that have been applied to zoos with references for their use in zoological collections.

Assessment Framework	Features	References and Species Examined
Five Domains	Criteria listed under 4 physical domains: Nutrition Environment Health Behavior Resultant 1 mental health domain	Multiple species [3]
European Welfare Quality^®^	4 principles: Good feeding Good housing Good health Appropriate behavior Principles defined by a set of 12 criteria	Bottlenose Dolphins [36] Dorcas Gazelles [37] Pygmy Blue-tongue Skink [10]
Universal Animal Welfare Framework	Four components: Institutional philosophy and policy Programmatic structure and resources Execution Evaluation.	None-overarching philosophy [16]
Opportunities to Thrive program	Flips the Five Freedoms to transform them to focus on attainment of positive affect: Opportunity for a well-balanced diet Opportunity to self-maintain Opportunity for optimal health Opportunity to express species-specific behavior Opportunities for choice and control	Hawaiian Endangered Birds-multiple species [38]
Animal Welfare Assessment Grid	Four components: Physical health Procedural Parameters Environmental comfort Psychological wellbeing	Zoo primates and birds [43] Giraffe, Scimitar horned oryx and large felids (tigers, leopards and cheetahs) [13] Western Lowland Gorillas [44]

## 5. Derivation of Welfare Indicators

Behavioral measures, once cross-validated, are becoming increasingly used in welfare assessments due to their practicality and non-invasive nature [11]. When selecting behaviors for inclusion, thought must be given to how species biology may influence assessment. When evaluating less expressive species, care needs to be taken to select measures that will give a clear representation of the animal’s welfare state. The same applies for species that may display seasonal differences in behavior, such as hibernation, or nocturnal species that may display different behaviors during the night when keepers are absent. The presence of abnormal behaviors is often included as a measure of welfare in behavioral-based assessment tools; however, caution should be taken. Whilst the presence of stereotypic behaviors may indeed indicate compromised animal welfare, these behaviors have also been shown to improve welfare when performed during stressful situations, and may persist in the absence of welfare compromise [15]. Their presence may therefore reflect historical rather than current welfare state [45] confounding interpretation of welfare scores.

Whilst it is common for assessment tools to score discrete behaviors, or groups of behaviors, that are widely considered to indicate a particular valence of affective state, for example play or abnormal behaviors, there are also a range of more formalized methods of deriving relevant indicators for inclusion in the developed tool. A number of these are described below.

### 5.1. Qualitative Behavior Assessment

Qualitative Behavior Assessment (QBA) is a method of summarizing and evaluating an animal’s behaviors and emotions in relation to their environment using descriptive terms such as ‘calm’, ‘timid’, or ‘excited’ [46]. It can be used retrospectively or applied to real-time welfare assessments. The method uses a comprehensive list of expressive terms, either generated by individual users when using the Free-Choice-Profiling methodology (FCP), or predetermined from the literature and expert consultancy. As FCP requires the participation of at least 10 observers and extensive data analysis [47], the latter is a more practical option for implementation in a zoo environment. QBA relies on the ability of keepers to observe subtle details of behavior, attitude, posture, and movement that may go undetected by other systematic behavioral data collection methods [11]. The literature shows good correlations between QBA and behavioral, physical, and physiological measures of welfare and studies have found high levels of inter-rater reliability when performed correctly by experienced keepers [11,47]. It is worth noting that whilst QBA is a useful resource to indicate the positive aspects of an individual animal’s welfare, it should not be used as a stand-alone assessment tool and cross validation with other measures is recommended [48].

In the context of animals in zoos, QBA has been used to develop a welfare assessment tool for captive elephants [49]. However, QBA was not used as the sole method of welfare assessment, and instead was combined with behavioral analysis of daytime and nighttime behaviors. This tool focused purely on behavior and did not incorporate measures of health and physical condition. Sixteen terms to describe demeanor were scored on a visual analog scale based on one-minute observation periods. The authors demonstrated that some QBA terms were able to be rated reliably by keepers and had an association with welfare state. However, some terms were unreliable; for example, the term ‘depressed’, intended to describe lethargy and disinterest in the environment, was interpreted by keepers in different ways, perhaps due to the common use of this terminology in relation to human mental health. As a result, it was suggested that ‘lethargic’ may have been a better descriptor. This finding implies that further validation of descriptors is required, along with their validation against alternative measures of affective state. Whilst these species-specific methods of welfare assessment may offer a more detailed indication of welfare than some generic assessment tools, the vast number of species held in any one zoo makes it impractical to use species-specific tools for all. This notwithstanding, the generic nature of QBA, along with the number of species for which its use has already been validated, would suggest that it could be adapted to a wide range of species housed in zoological parks.

### 5.2. Behavioral Diversity

In recent years there has been a growing interest in the use of behavioral diversity as a measure of animal welfare. Behavioral diversity is defined as the number of behaviors, as well as the frequency of each behavior [50]. The assumption made is that when diversity is high, behavioral needs are being met, but when diversity is low an animal may be in poor welfare; for example, they may show stereotypies or be lethargic [50]. The Shannon–Wiener Diversity Index (H-index) is perhaps the most used measure of diversity and provides an indication of whether an animal’s time budget is mostly made up of a few behaviors or covers a range of behavioral categories [51]. The model closely aligns with the natural living model of welfare, in that animals in the wild display considerable behavioral diversity; hence, good welfare in captivity would be assumed if animals showed similarly diverse behaviors [23]. However, diversity indexes require counting of different individual behaviors whereas the natural living model of welfare compares behaviors that are similar between wild and captive counterparts. Therefore, if the assumption that increased diversity correlates with good welfare holds, a valid welfare assessment method would be comparison of time budgets between captive animals and their wild counterparts [52,53]. Alternately, ethogram comparisons between before and after environments or conditions would allow assessment of welfare status change [53]. However, there is a risk of arriving at incorrect assumptions about welfare. For example, the performance of behaviors associated with negative affective states may increase, causing a change in diversity index, but the welfare state may have declined [54]. For this reason, the inclusion of so-called ‘negative’ behaviors is needed to avoid misinterpretation.

In spite of these concerns, there is a growing attention to behavioral diversity as a valid positive indicator of welfare. Firstly, there is now an extensive body of evidence showing that management practices that are thought to improve welfare lead to increased diversity. For example, the impact of enrichment or animal training on greater behavior expression has been widely reported (see [50] for review). It is also promising that this change is occurring across orders and species and is not confined to mammals alone. Secondly, there appears to be a relationship between diversity and physiological indicators of welfare, for example with corticosterone as a measure of HPA axis activity with the expected inverse relationships being found in cheetahs [55] and dolphins [56].

In practice the method probably has greatest application via use of an established diversity index prior to and following a husbandry change, rather than as a one-off assessment method. Monitoring of behaviors can be achieved through targeted focal animal sampling, with the use of technology to aid practicality, e.g., the use of the Zoo Monitor program. However, as described by Miller et al., 2020 [50], use of the method may require a shift in focus away from keeper-oriented husbandry programs with predictable scheduling to opportunity-based changes considering the behavioral outcomes observed. This paradigm shift would require whole-scale institutional endorsement and commitment to optimize success.

### 5.3. Cognitive Bias Assessments

Assessing what is important to animals and how they respond to certain cues can be a useful tool in measuring affective states. One process that has been applied to captive animals in an effort to better understand how emotional states can affect cognitive processes is cognitive bias testing. This methodology is originally from human psychology research and uses classical or operant conditioning to measure an animal’s response to ambiguous cues. The premise is that in tests an animal with an “optimistic” bias will react more positively to neutral stimuli and vice versa [13,57]. Clegg, 2018 provides a review of its use in zoos, reporting across many species and contexts that an animal’s cognitive bias can be linked to welfare state, e.g., those in better welfare make more optimistic judgements. Although this methodology has considerable potential to allow a more comprehensive understanding of affective states in zoo animals, there are practical limitations in its application as it requires some experimental set up and resources dedicated to animal training [57].

### 5.4. Delphi Consensus Methods

Whilst not a welfare assessment technique, use of expert opinion can be a useful method for gathering information to feed into welfare assessment or welfare planning tools. For example, in cases of limited species literature on welfare indicators, the method can be used to establish consensus by formalizing a communication process to source this information from experts [58]. The technique usually comprises at least two rounds of questionnaires or interviews, with an iterative approach such that results from the first round are reconciled for the next round for further questioning of participants [59]. The method has been used to identify welfare indicators in a range of species commonly housed in zoos including elephants [60], tigers [61], and reptiles [62]. One application of this methodology is the development of an Animal Welfare Priority Identification System© (APWIS©) [60]. This Delphi-based process was trialed to determine the welfare significance of individual behaviors and cognitive processes for Asian elephants. The system examines the motivational characteristics, evolutionary significance and likely welfare impacts of individual behaviors and cognitive processes. The results indicate that species-specific social and cognitive opportunities are important to the welfare of Asian elephants in captivity and as such, this knowledge should be used to inform husbandry guidelines, habitat design, management strategies, and animal welfare assessments [60].

As the method relies solely on opinion (albeit expert), it is not possible to confirm linkage of indicators with valence of affect. However, confirmatory analysis of behaviors identified can be performed in later follow-on studies. Furthermore, the participatory nature of the method likely encourages communication and greater buy-in to the tools that arise by staff at zoological parks and their institutions.

## 6. Important Features of Welfare Assessment Tools for Zoos

An individual animal’s welfare state is multifactorial and as further understanding evolves, the importance of including animal-based measures in welfare assessments has been highlighted [11,22]. There is a consensus among researchers that in addition to ensuring all resource needs are met, positive affective states should be promoted. There are many factors to consider when developing a welfare assessment tool and determining how welfare should be measured and recorded can be a challenge [11,13,63]. The various conceptual frameworks have given rise to multiple tools, but these are often developed for specific species or contexts and there is no ‘one size fits all’ [64]. Under captive conditions, such as those seen in zoos and aquaria, this variation may result in non-comparable data limiting the ability for knowledge-sharing and collaboration between organizations. Additionally, Fraser 2009 noted that the assessment of welfare using a tool created based on a single criterion may produce a poor welfare result when scored by a tool based on different criteria [65]. As a result, there is a requirement for standardized guidelines to developing zoo animal welfare assessment tools that are adaptable and flexible enough to suit various taxon or species in varying contexts, whilst still producing comparable datasets. The development of such guidelines would need to consider features such as validity and reliability and the split between animal and resource-based assessments. They would also need to consider factors such as seasonality, highly motivated innate behaviors, social settings, and species management requirements such as breeding recommendations. In addition, scoring schemes, the weighting of criteria, and methods of determining criteria should be regarded.

### Tool Features: Validity, Reliability, Practicality

A welfare assessment tool should be practical, reliable, and simple enough to allow completion by observers with no specialist training, as well as being comprehensive enough to allow application to a variety of species, at least within taxa, to enhance practicability. A combination of multiple measures should be used to determine an animal’s welfare status. Prior to inclusion in an assessment tool, these measures need to be evaluated for practicality (how easily they can be observed), reliability (the consistency in which they can be assessed) and validity (the ability of the measurement to reflect the desired construct) [49]. It has been suggested by Yon and colleagues [38] that a fully validated assessment tool should evaluate the various types of validity and reliability against predetermined thresholds from the literature. Common types of reliability applied to welfare assessment include inter-/intra-rater, test re-test reliability, and internal consistency. Validity may refer to content, concurrent criterion or known groups criterion validity [49].

When developing a welfare assessment tool, the recommended frequency of implementation is another factor that must be considered in order for it to remain practical. If recommending daily use by keepers, the tool must be rapid and simple to complete. More thorough institutional assessment tools may need less frequent evaluations. If the data are to be collected over the long-term to chart trends, consideration of reliability across time incorporating aspects of inter and intra-observer variability is also important, as well as the influence of animal-based indicators on scoring (see below).

The training and species knowledge of keepers may also influence the type of measures and methods selected for inclusion in the assessment tool. Multiple studies have validated keeper assessments as a measure of welfare by correlating observations with other common physiological and behavioral indicators of welfare [42,66]. However, these types of assessment require observations by experienced caretakers. There has also been suggestion that keepers tend to capture more subtle changes to welfare than researchers using retrospective analysis, although correlation between different methods of assessing welfare, such as behavioral observations versus general welfare indicators may not be strong [44]. Conversely, Wemelsfelder and colleagues found that when using QBAs, inter-rater reliability was high even when performed by observers with no previous experience with the species [46,67,68].

As a further consideration there is the potential for keeper interpretations to be influenced by an emotional attachment to the animals under their care. This may be more likely to occur where welfare is reduced, such as prompting an overly positive welfare assessment to seek to avoid euthanasia being considered for a beloved animal in the face of a chronic unresolvable health concern. Conversely, that emotional attachment can enhance a keeper’s resolve to not see the animal under their care suffer, and to actively approach euthanasia as a positive welfare outcome in such a circumstance.

Validating measures of welfare can be challenging in a zoo environment due to the often-small sample sizes available and the limited literature on natural behavioral biology for many species [13]. However, there are numerous ways in which criteria can be selected for inclusion. A common approach includes a combination of reviewing the existing literature, gaining opinions through expert consultancy, and then modifying previously established and validated assessment tools. When developing a novel welfare assessment tool for captive elephants, Yon and colleagues conducted a review of both peer- and non-peer-reviewed literature and consulted stakeholder focus groups to identify novel measures for inclusion [49]. In their 2018 Assessment of Welfare in Zoo Animals, Wolfensohn and colleagues recommended the use of a targeted version of the Animal Welfare Assessment Grid (AWAG) to create a ‘gold standard’ welfare improvement benchmark for zoos over time [13]. AWAG is an easily adapted tool that has been previously validated in primates [69] and gorillas [44] and successfully adapted for use in zoos to highlight welfare impacts [43]. In addition, the AWAG system has been trialed on zoo collections of giraffe, scimitar horned oryx, and numerous large felids [13].

Conducting direct behavioral observations and welfare assessments can be time consuming and limited to individuals with specialized training. Recent efforts have been directed towards making these processes more efficient for caretakers through the use of supportive monitoring technology. Diana and colleagues (2021) conducted a systematic review of the literature that has explored the use of technology to monitor zoo animal welfare with a focus on real-time automated monitoring systems [70]. They found 19 publications that have investigated the use of sensor technology including wearable devices or automated monitoring systems embedded in enclosures to measure animal behavior. The application of such remote monitoring tools in zoos is in its infancy, but there is potential for this approach to evolve into a more useful and accessible tool for zoos to employ to support animal welfare monitoring.

Another application of technology for zoos is to support data collection and management. Lincoln Park Zoo developed a systematic behavioral monitoring app called ZooMonitor to provide a low-cost and user-friendly tool for zoo staff to conduct behavioral monitoring. Zoo staff and researchers can use ZooMonitor to record behavior and habitat use and log individual animal characteristics such as body condition, with the data uploaded to a cloud server allowing the user to conduct automated reliability tests and generate reports including heat maps of space use and activity budgets [71]. The Zoological Information Management System (ZIMS) by Species360 [20] is a global database for zoo and aquarium members to manage animal records, and includes an animal care and welfare module. This module is designed to streamline access to welfare indicators for users. Care and welfare indicators are provided globally but each institution can define the parameters and assign expected values or value ranges for each indicator at a taxonomic level. Although these tools still require time dedicated to behavioral observations, the use of technology can make the data collection, reporting, and summarizing phases more efficient.

## 7. Factors for Consideration in Development of Welfare Assessment Tools

### 7.1. Animal-Based Considerations

There are a variety of animal-based considerations that influence behavior and the animal’s affective experiences and consequent welfare. For example, sex differences, nutritional requirements, life history events, social structure, natural behaviors, and individual personalities may all need consideration in developing welfare assessment tools (see [72] for extended discussion on these individual factors)

Across the life cycle of a species there are key considerations that are important to consider when assessing welfare. As a result, resources may need to differ as a result of age. For example, juveniles need space and time for play and opportunities for social learning. In animals of reproductive age, closer attention needs to be paid to social structures and ability to protect young (see [72] for further examples). Assessment tools may then have to be tailored to age of animals. Animals of different ages may display varied responses to the same situation or provision of resources. This is likely representative of a situation having varied impacts on welfare of animals at differing stages of physiological and neurological development [73]. Having baseline data for particular age groups for later comparison will aid in creating a reliable tool.

Significant variations in behavior of zoo animals based on time of day and season are documented in the literature. Animals can be categorized based on their activity patterns, such as diurnal, nocturnal, crepuscular (dawn and dusk-active), amongst others. Since caretakers are usually only present during daytime hours, it is important to consider how this may impact recording of animals’ activity budgets. As an example, animals that naturally forage at night may have to wait for food according to the keeper’s schedule. A range of husbandry techniques may be used in zoos to address such issues, including reverse cycle light systems and devices to facilitate delayed release of food items. It has been noted that blue lights used in reverse cycle light systems may negatively impact activity and health [72]. Seasonality is evident in many captive species including African elephants (*Loxodonta africana*) [74], grizzly bears (*Ursus arctos horribilis*) [75], Rothschild’s giraffes (*Giraffa camelopardalis rothschildi*) [76], and ring-tailed lemurs (*Lemur catta*) [77], and this needs to be factored in when using behavioral measures as indicators of welfare. For example, in some species, inactivity may be observed more frequently during winter due to innate hibernation behaviors [75] while other species may become more active [78]. Some animals might spend less time feeding and more time lying down during the warmer months [74]. Many species show seasonal reproductive patterns [79] which may influence the frequency and type of social interactions observed. Thus, it is important that assessments are comprehensive enough to account for seasonal variability, and that evaluations of such behaviors do not produce a misleading negative partial welfare score. For this reason, there is increasing merit in the inclusion of keeper records in welfare assessments as caretakers are often most familiar with the individual temperaments, personalities, and behaviors of the animals in their care [8].

The relationship between captive animals and humans has been identified as a key contributing factor to the way in which animals interact with their environment [80]. An animal’s level of exposure to training or to less formal human contact may act as a modifier of behavior in that the individual may become more confident when interacting with humans. It is inferred that an increase in positive human–animal interactions (HAR) translates to an improved welfare state [80]. Therefore, animals that have received greater exposure to humans are likely to demonstrate different responses in their reactions to observers than their most extensively managed counterparts, potentially influencing welfare assessment scoring if these are not able to be performed remotely. Similarly, animals’ previous experiences or familiarity with their enclosure space and conspecifics will influence the behaviors observed [81]. For this reason, if animals have recently experienced an enclosure or social group change it may be prudent to perform formal welfare assessment scoring once a period of habituation has been completed to allow accurate comparison between scores across time.

There is diversity in the purpose of animals held in zoos and aquariums, and the associated variations in management may require consideration for welfare assessment. Animals may be permanently under captive management as part of long-term insurance populations, be part of reintroduction programs for release into conservation areas, or be wild-caught animals under temporary captive management for a range of reasons. They may be managed in breeding situations or held as non-breeding groups, and may be managed extensively with minimal human interaction through to intensive conditioning to human presence for encounter and presentation purposes. For reintroduction programs, the management practices required to optimize the survival of animals post-release often vary substantially from the approach taken to animals in permanent captive care. Welfare assessment strategies for animals in release programs may need to be tailored to focus on those outputs which are most likely to maximize survivability in the wild [38]. For example, foraging competency may be essential for survival in the wild and hence this should be assessed prior to release.

A further consideration is that animals may differ in how they come to live at zoos, and as a result have varied prior life experiences. Zoos may acquire animals from non-zoo private sources, wild situations under permit, or may rehabilitate sick/injured wildlife [82], whereas other animals may have been born into the zoo environment. Origin of animals (wild versus captive-born) has been demonstrated to alter animal-based parameters which may be included as welfare assessment criteria. For example being captive-born modified behavioral expression in black rhinoceros (diceros bicornis), particularly in relation to human–animal interactions [83], led to reduced lifespan in Asian elephants [84], and was linked with reduced behavioral organization and increased stereotypy formation in bears [85]. An awareness of this when contrasting scores across individuals is needed.

### 7.2. Scoring Methodology

There are a variety of considerations in relation to scoring schemes for assessment tools. Users have to be clear on the purpose of the assessment and how the data will be used. For example, where the data are being used to prioritize species groups or enclosures where further research or investment is needed, absolute score values may be less important than how these areas rank with other competing priority areas. However, if the aim is to assess individual animal welfare, then absolute scores and the interpretation of how these reflect valence of affective state are critical. It is critical that the scheme used is standardized to the greatest extent possible and applied consistently each time the welfare-assessment tool is used.

In order for an assessment tool to be valuable it must be able to record changes in welfare status over time. This can be achieved by generating an objective welfare score which can be used for comparison in future evaluations. These scores are also used by accreditation schemes to determine if an organization is meeting welfare requirements. Numerical scales are most common as they allow for ease of data analysis and comparison between datasets. Sherwen and colleagues implemented a 0–2 scale for risk of compromised welfare (where 0 represents high risk, 1 represents a moderate risk, and 2 represents no observable risk) when developing their Animal Welfare Risk Assessment Process for Zoos [3]. A three-point scoring system was also used in a modified version of the Welfare Quality^®^ Protocol to assess welfare in the Pygmy Blue-Tongue Skink, where assessment against the descriptor resulted in a determination from poor to good [10]. Likert scales are also commonly utilized when evaluating behavioral indicators. When asking observers to score daytime behavioral frequencies in captive elephants, Yon and colleagues provided a variety of Likert scales with responses ranging from ‘never’ to ’more than once per day’ where appropriate, and utilized various numbers of response options based on the expected frequency of that behavior [49]. A simpler alternative is to use a checklist approach where each measure is compared against a minimal requirement yielding a yes/no response. This method is simple, quick to use, and easy to standardize, and may be the most appropriate method when determining legislative compliance. However, checklists lack sensitivity and rely on the assumption that all measures are of equal importance. As a result they are less useful for comparing between zoos or across groups of animals to determine need for resource allocation [86].

When combining results of welfare measures to gain an overall assessment score, there are a number of ways in which criteria may be weighted [86]. Delphi surveys or similar methods of expert consultancy, such as those employed by Veasey 2020 in an attempt to identify the psychological priorities of captive elephants, are an informal method of aggregating results to derive weighting based on validity, practicality, and expected frequencies [60]. Alternatively, precise calculations can be used; the sum (or mean) of scores is commonly used since the concept can be easily grasped by a wide audience. In this method, all indicators are ideally first converted to a unified numerical scale (e.g., 0–2 scale used by Sherwen and colleagues [3]) before weightings are assigned. Weightings are assigned based on the impact of the measure on animal welfare such that those with reduced validity are assigned lower weightings [86]. It is important to note that if a unified numerical scale is not implemented, the scale and intervals between levels for each measure should be illustrated to avoid confusion or misrepresentation. By combining the weighted sum of the scores an overall welfare score can be calculated. There are, however, some limitations to the use of this method. Scales and intervals between levels are often assumed, not illustrated, resulting in confusing or misleading results; the nature of the method allows for full compensation, i.e., a major welfare compromise may be neutralized by multiple minor advantages; and the method cannot favor situations of compromise (see [35] for full description). The sum or mean of ranks has been used in farm animal assurance schemes as a method of deriving an overall assessment. However, this method is only of use when comparing enterprises. Farms are ranked on each measure, a mean of all of these ranks calculated, and farms sorted based on their overall rank [87]. Scores may also be derived relative to a standard comparator. For example, in Brouwers and Duchateau’s 2021 study using the AWAG (see above) for assessing gorilla welfare, indicators were scored relative to a healthy individual of the same sex and age, on a ten-point scale with one as the optimum [44]. An interesting visual method of presenting scores (also see [43]) is plotting average scores, across the indicators, in the four selected physical, procedural, environmental, and psychological domains on a radar plot. This allows calculation of a Cumulative Welfare Assessment Score (CWAS) based on the surface area of this chart (see Figure 1). As a result, the CWAS increases exponentially rather than linearly when different classes are affected. This aids in capture of long-term trends in welfare [44]. These radar plots can also be created separately to visualize group differences, for example having a plot for all-male gorillas versus the family group. Having standardized scoring systems for zoos and aquaria would allow for ease of data comparison between organizations, allowing zoo managers to create a ‘gold standard’ which all organizations should strive to achieve.

## 8. Application

The focus of this review was to outline the basis for welfare assessment frameworks that are in use or have been trialed in zoos with a focus on the foundation for the criteria derived, and to explore the method of deriving a practical and useable tool based on these foundational principles. Based on the discussion above, a summary flowchart of considerations in development and utilization of a welfare assessment tool is proposed in Figure 2. The thinking and steps outlined in Figure 2 have not been empirically tested, and there remains a future need to evaluate the effectiveness of this process across different species and tool types. To illustrate the process Table 2 presents a case study of how these steps and considerations might operate in practice based on the authors’ experience of developing a welfare assessment tool for the pygmy blue-tongue skink (*Tiliqua adelaidensis*). Using these methods is likely to lend structure to the tool derivation process and assist in highlighting the reflective and iterative processes involved.

## 9. Conclusions

With increasing interest from both the public and from within institutions themselves in continually improving the welfare of animals held in zoos, the use of tools to make evidence-based assessments of zoo animal welfare is crucial. Details of how to accomplish this are not always specified by accreditors, and thus the establishment of standardized guidelines for developing zoo animal welfare assessment tools would be beneficial for data comparison and sharing. In this review we have proposed a process for engaging with this task. This process builds on the structure provided by adaptation of existing welfare assessment frameworks, but highlights the ongoing iterative modifications needed as evidence and experience is gained.

## Figures and Tables

**Figure 1 vetsci-09-00170-f001:**
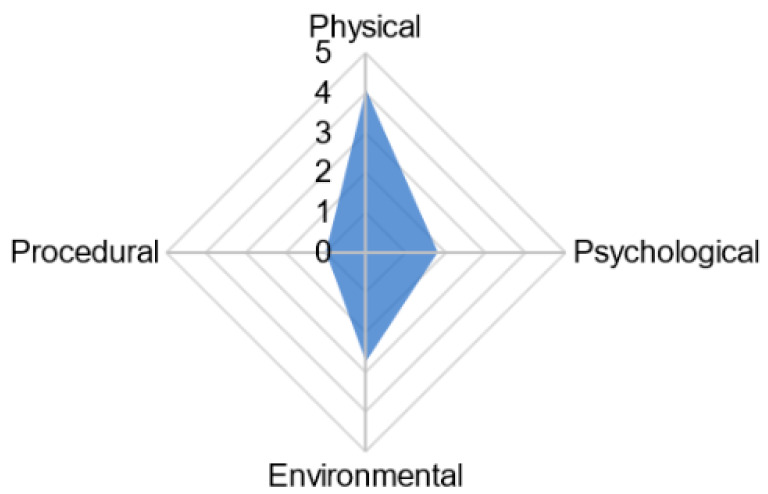
Radar chart illustrating visualization of animal welfare data by using averages of the four parameter classes of physical, procedural, environmental, and psychological to form a polygon. The CWAS is represented by the surface area of this chart. Reproduced from Brouwers, Stijn and Duchateau, Marie José 2021, “Feasibility and validity of the Animal Welfare Assessment Grid to monitor the welfare of zoo-housed gorillas Gorilla gorilla gorilla”. Journal of Zoo and Aquarium Research 9(4):208. Reprinted under a Creative Commons Attribution CC BY 4.0 license.

**Figure 2 vetsci-09-00170-f002:**
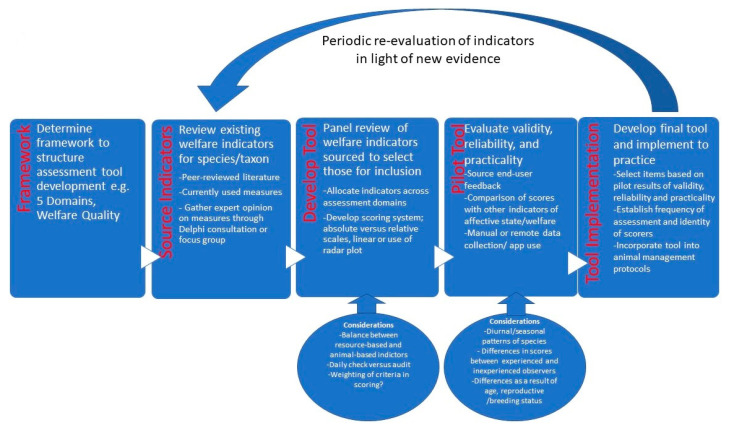
Summary flowchart of steps and considerations involved in deriving a welfare assessment tool for use in zoos.

**Table 2 vetsci-09-00170-t002:** Illustrated example of the assessment tool derivation process based on a case study with the pygmy blue-tongue skink (PBTS) (*Tiliqua adelaidensis*) (adapted from [10] under a Creative Commons Attribution CC BY 4.0 license).

Step in Tool Derivation Process	Process Performed	Outcome of Step	Examples of Considerations
Framework Choice	Contrasted frameworks available and selected Welfare Quality Protocol due to clear guidance on criteria provided	Selection of Welfare Quality Protocol	Frameworks considered were Five Domains and Welfare Quality due to previous experience with use
Source Indicators	Species- relevant criteria were derived by local consultation with keepers and veterinarians, and review of broader literature on lizards	Derived 39 animal or resource-based indicators, aligned against the 12 Welfare Quality criteria and 4 principles	Animal cleanliness was derived as a measure of comfort around resting- the literature suggests that scat piling may be disturbed if welfare poor [88]
			The PBTS is omnivorous, and their diet should consist of 50% vegetables, 25% fruits, and 25% invertebrates such as snails. This needs consideration under the criteria “appropriate diet” for the category of ‘good feeding’ [89]
Review of indicators to determine those for inclusion	Review of derived indicators by team comprising zoo veterinarian, researchers, animal welfare officer and two reptile keepers	Removed one animal-based indicator to yield 38 indicators. Tool was made up of predominately animal-based indicators (77%) and was designed to be	Tail autotomy was initially identified but removed since in the particular genus of interest, Tiliqua, there are reduced planes and tail drop is therefore unlikely [90]
		part of a longer audit-type assessment.	Tool primarily comprised animal-based indicators as suggested in EU Welfare Quality documentation
		Adapted grading system from Sherwen at al. 2018 [3] with scoring for from 0–2 representing high risk to low risk for resource-based indicators, and poor to good welfare for animal-based measures. The overall score was determined by summation and determination of percent out of the maximum score possible.	
	Scores were not weighed since there was no evidence at hand to determine relative importance of the criteria in terms of indicators of animal welfare	
Evaluate validity, reliability and practicality	Pilot study performed on a breeding pair of pygmy blue-tongue skinks and their enclosure through manual observation	Observation took 2 h for this pair. This is likely suitable for an audit-type assessment but would have been excessive for a daily check	Noted that some criteria could not be observed but were not necessarily a sign of compromised welfare. For example, food was not presented at the time of observation so food intake and hunting behaviour could not be assessed. This highlighted the need to consider incorporating information from records since these criteria had been observed previously the same week.
		Observation performed in winter	Understanding that due to ectothermic physiology and lizards being dormant, behavior may differ in contrast to summer assessment. In spite of this animals scored 79% implying good welfare (based on a 60% threshold).
Develop Final Tool and Implement	Review of derived indicators by team comprising zoo veterinarian, researchers and reptile keepers	Three resource-based indicators were added following observations to yield tool with 41 indicators	Three resource-based indicators were added; enclosure cleanliness, maintenance and group size
		Ensure welfare assessment done at a time when food available so the feeding- related criteria can be assessed	After reflective process, decided to continue work to expand the derived tool out to other reptiles with a focus at taxa level
	Determine identity of scorers and that tool most appropriate for audit-style assessment	Implement and continue to refine tool based on feedback from users and considering corroboration with other indicators e.g., information from health records.

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
