# Peer review of "Welfare Assessment Tools in Zoos: From Theory to Practice"

_vetsci, 2022, doi:10.3390/vetsci9040170_

Round 1

Reviewer 1 Report

This is a generally well-written summary of the existing animal welfare assessment tools and frameworks in existence, including the theoretical application of these to zoo animals. However, there is too little critical discussion of each of these aspects to classify as a critical review and there is lack of adherence to PRISMA guidelines to classify as a systematic review. The manuscript itself does not attempt to clarify what type of review this is, although the term "comprehensive review" was used in the abstract. There was no description of how the publications included in this "review" was chosen, e.g. what search terms, databases etc. were used. As such, I do not believe that this is an appropriate manuscript submitted as a review article to this journal. It may be more appropriate as a "commentary" or "opinion" type manuscript in a journal that accepts such articles.

If this were to be assessed based on the merit of contents and discounting its suitability as a review article, the authors specifically mentioned that they "propose a ‘how-to’ guide to creating welfare assessment tools for zoological parks". For this particular purpose, the manuscript is also not an easy to use 'how-to' guide. The summary of the existing frameworks in Table 1 does not lead on to an explanation of how these frameworks can be used in creating welfare assessment tools nor a discussion on the species each framework can be applied to. There are some useful pointers and anecdotes scattered throughout the different sections that remind a reader on the important considerations when formulating animal welfare assessments, but a few case studies representing a few different species to demonstrate how animal welfare assessments can differ between species in terms of the tools applied and the implementation of these assessments would make this a much more useful practical 'how-to' guide.

I believe that there are important points raised in this manuscript, but the current way in which it is presented makes it neither suitable as a review (as it is submitted for) nor a 'how-to' guide (as it is presented in the introduction). Therefore, I believe that significant reorganization of the content, as well as addition of suitable case studies, will be required before this can be considered for publication in a more suitable section and/or journal.

Author Response

This is a generally well-written summary of the existing animal welfare assessment tools and frameworks in existence, including the theoretical application of these to zoo animals. However, there is too little critical discussion of each of these aspects to classify as a critical review and there is lack of adherence to PRISMA guidelines to classify as a systematic review. The manuscript itself does not attempt to clarify what type of review this is, although the term "comprehensive review" was used in the abstract. There was no description of how the publications included in this "review" was chosen, e.g. what search terms, databases etc. were used. As such, I do not believe that this is an appropriate manuscript submitted as a review article to this journal. It may be more appropriate as a "commentary" or "opinion" type manuscript in a journal that accepts such articles.

The manuscript is presented as a traditional narrative review, which tend not to add specific search terms since as the reviewer points out no systematic process in deriving these was used. However, we do believe we have captured the majority of the publications in this area, and of course its a relatively small study field. Search terms can be added if required but from our perspective it seems to provide less value since the non-systematic nature means we do not expect anyone to replicate the review.

A systematic review on this area would have to be a scoping review in format since there is no clear 'intervention' question to fit the PICO criteria, and would differ very little in format from that presented here except for the presentation of the search criteria and inclusion of the PRISMA (scoping) flow diagram. Scoping reviews still at their heart though require a focussed question and our idea for this review is probably a little broad to fit this type of review. 

It is our opinion that the review is entirely scholarly literature-based so cannot be validly classed as our 'opinion', although a commentary is a possibility. 

We believe that the literature has been critiqued within the text, placing it within the broader contribution and stating alternative theories/methods to the proposition where they exist. 

If this were to be assessed based on the merit of contents and discounting its suitability as a review article, the authors specifically mentioned that they "propose a ‘how-to’ guide to creating welfare assessment tools for zoological parks". For this particular purpose, the manuscript is also not an easy to use 'how-to' guide. The summary of the existing frameworks in Table 1 does not lead on to an explanation of how these frameworks can be used in creating welfare assessment tools nor a discussion on the species each framework can be applied to. There are some useful pointers and anecdotes scattered throughout the different sections that remind a reader on the important considerations when formulating animal welfare assessments, but a few case studies representing a few different species to demonstrate how animal welfare assessments can differ between species in terms of the tools applied and the implementation of these assessments would make this a much more useful practical 'how-to' guide.

I believe that there are important points raised in this manuscript, but the current way in which it is presented makes it neither suitable as a review (as it is submitted for) nor a 'how-to' guide (as it is presented in the introduction). Therefore, I believe that significant reorganization of the content, as well as addition of suitable case studies, will be required before this can be considered for publication in a more suitable section and/or journal.

To increase the practical utility of the review, we have included a flowchart (Fig 2) of the steps involved in creating a  welfare assessment tool. We believe that it would not do justice to the included papers in the review to present them as case studies, since these case studies are the basis for a whole peer-reviewed journal article. References to previous studies developing assessment tools are throughout this work and highlighted particularly in Table 1. 

Reviewer 2 Report

The paper submitted by Jones et al. aims to summarise different methods to assess welfare in zoo-housed animals. The manuscript is very detailed, and can give a good overview over the different methods and can be used as a guideline when planning a welfare assessment. However, it would be easier to implement, if the authors would give some more examples of studies on different species (in cases where it is possible) and to go into more details of the pros- and cons of the different methods.

Minor comments

L 14 Please add a full stop after measures

L143 & L 161 Please add years of publication as you did in the following paragraphs.

L 387 Please add: …can ‘be’ used…

Author Response

The paper submitted by Jones et al. aims to summarise different methods to assess welfare in zoo-housed animals. The manuscript is very detailed, and can give a good overview over the different methods and can be used as a guideline when planning a welfare assessment. However, it would be easier to implement, if the authors would give some more examples of studies on different species (in cases where it is possible) and to go into more details of the pros- and cons of the different methods.

We do understand the concern but feel like we could not really do justice to the published case examples out there if we try and cram them into this kind of review. Authors tend to publish them as stand-alone papers and many of them (if not all) are referenced in this document. However, in order to improve the utility of the review we have included a flowchart (Fig 2) on steps to derive a welfare assessment tool. We think this is a beneficial addition, providing a quick glance summary.

Minor comments

L 14 Please add a full stop after measures

This has been added. 

L143 & L 161 Please add years of publication as you did in the following paragraphs.

Corrected. 

L 387 Please add: …can ‘be’ used…

Corrected. 

Reviewer 3 Report

This is an overall well written and comprehensive review that outlines the basis for welfare assessment frameworks in use or trialed in zoos, with a focus on the foundation for used criteria. The authors have done an excellent job in summarizing and assessing available literature and to explore the method of deriving a practical and useable tool based on foundational principles. As they rightfully state, increasing interest from both the general public and from within institutions themselves warrants such a review, which will aid the continual improvement of welfare of captive animals. The review highlights the need to establish taxon or species- specific assessment tools across all taxonomic groups to inform welfare management strategies.

A few minor changes are recommended below.

Overall:

I recommend to refrain from using “we” (in reference to the authors themselves) throughout the manuscript. 

Try to reduce the use of “also” and avoid unnecessary filler words such as “certainly”, “as such” or “perhaps” throughout the manuscript.

Minor changes

Line 14 – fullstop missing.

Line 29 – Assume „not“ needs to be removed here (correct to: is not justified unless high standards of animal welfare are apparent)

Line 59 – remove “furthermore”

Line 87 – put animal welfare in inverted comas as has been done in line 88

Line 125 onwards

Another important point to consider adding – displaying signs of discomfort, stress or illness is an “unnatural” response for many species. In many species, it has the potential to lead to a loss of social status or standing within a group or can result in the animal being targeted as pray. Therefore, animals will refrain from displaying these signs for as long as possible.

Lines 244-259

Some additional thoughts here: whilst, as the authors report, keepers are generally excellent in recognizing changes in animal’s behavior and condition, the emotional attachment many form to their subjects of care can result in a degree of clouded judgment, in particular where assessment of reduced welfare is concerned, especially relating to time points of euthanasia in chronically ill individuals.

Line 290

Remove commas after “whilst”, and “behaviors”

Line 291

Remove comma after “behaviors”

Author Response

This is an overall well written and comprehensive review that outlines the basis for welfare assessment frameworks in use or trialed in zoos, with a focus on the foundation for used criteria. The authors have done an excellent job in summarizing and assessing available literature and to explore the method of deriving a practical and useable tool based on foundational principles. As they rightfully state, increasing interest from both the general public and from within institutions themselves warrants such a review, which will aid the continual improvement of welfare of captive animals. The review highlights the need to establish taxon or species- specific assessment tools across all taxonomic groups to inform welfare management strategies.

A few minor changes are recommended below.

 Overall:

I recommend to refrain from using “we” (in reference to the authors themselves) throughout the manuscript. 

I can only find one use of ‘we’ which is in the introductory paragraph. This paragraph has been modified slightly in response to reviewer feedback but we have put this sentence into the passive tense now.

Try to reduce the use of “also” and avoid unnecessary filler words such as “certainly”, “as such” or “perhaps” throughout the manuscript.

 Where we have caught these filler words they have been deleted.

Minor changes

Line 14 – fullstop missing.

This has been amended.

Line 29 – Assume „not“ needs to be removed here (correct to: is not justified unless high standards of animal welfare are apparent)

Thankyou- the second ‘not’ has been removed.

Line 59 – remove “furthermore”

Furthermore at the beginning of the paragraph has been removed.

Line 87 – put animal welfare in inverted comas as has been done in line 88

Inverted commas have been added at line 87.

Line 125 onwards

Another important point to consider adding – displaying signs of discomfort, stress or illness is an “unnatural” response for many species. In many species, it has the potential to lead to a loss of social status or standing within a group or can result in the animal being targeted as pray. Therefore, animals will refrain from displaying these signs for as long as possible.

I have added to the paragraph at line 125 the following: “The latter may be particularly important since displaying signs of stress or illness may lead to a loss of social standing within a group, or could result in the animal being targeted as prey.”

This point is one of interest to me and I have been challenged on this previously in review of papers with a reviewer suggesting there is no empirical evidence of this assertion. I must admit I am unable to find a primary citation supporting this but it is certainly a widespread assumption held by the veterinary, biological and pain scientist community, and one which I have held.  The statement as expressed I hope conveys that this is a hypothesis rather than as established fact based on the use of the term ‘may’.

Lines 244-259

Some additional thoughts here: whilst, as the authors report, keepers are generally excellent in recognizing changes in animal’s behavior and condition, the emotional attachment many form to their subjects of care can result in a degree of clouded judgment, in particular where assessment of reduced welfare is concerned, especially relating to time points of euthanasia in chronically ill individuals.

Thankyou for this suggestion. We agree this is certainly a possibility and have added referral to it in section 5.1. 

 Line 290

Remove commas after “whilst”, and “behaviors”

We have removed the comma after whilst. We feel that the other commas are necessary.

Line 291

Remove comma after “behaviors”

We feel that the comma is needed here.

Round 2

Reviewer 1 Report

I appreciate the acknowledgement that this is a traditional narrative review, but the term "comprehensive review" is still in the abstract, which can be misleading. I would also recommend including in the "introduction" the explanation provided in the "response to reviewer" regarding the type of review this is supposed to be (albeit not ad verbatim, but modified to flow with the manuscript).

Figure 2 does add some practicality to the content of the text, as does the additions to various parts of the manuscript. However, my previous recommendation for more case studies was not in relation to the papers that were already cited, but rather to apply the principles summarized in the manuscript on real-world examples of how these principles were used on animals in the authors' institutions. Without these type of examples, the practicality of the manuscript is still limited.

Author Response

I appreciate the acknowledgement that this is a traditional narrative review, but the term "comprehensive review" is still in the abstract, which can be misleading. I would also recommend including in the "introduction" the explanation provided in the "response to reviewer" regarding the type of review this is supposed to be (albeit not ad verbatim, but modified to flow with the manuscript).

We have removed the terminology 'comprehensive' review in the abstract and inserted 'narrative' before review in the last paragraph of the introduction. We were concerned that the flow of the manuscript would be disrupted by including details on the type of review performed in the introduction. Instead we have included another section (number 2) headed review methodology and included some basic details of the review search methods. This section reads:' 

The review methodology was not systematic in nature given the breadth of the review question with no clear intervention or outcome criteria. However, structured searches were performed in the following databases: CAB Abstracts, Web of Science and Scopus. Keyword search terms used included: ‘zoo’ AND ‘welfare assessment’, ‘captive animal’ AND ‘welfare’, ’zoo’ AND ‘well-being’, ‘zoo’ AND ‘welfare tool*’, ‘welfare tool’ AND ‘scoring’. In addition, the reference lists from sourced papers were scanned for additional citations, and forward citation searching was performed on papers identified.

Figure 2 does add some practicality to the content of the text, as does the additions to various parts of the manuscript. However, my previous recommendation for more case studies was not in relation to the papers that were already cited, but rather to apply the principles summarized in the manuscript on real-world examples of how these principles were used on animals in the authors' institutions. Without these type of examples, the practicality of the manuscript is still limited·      

Thankyou for your thoughts on this and we do of course want the review to be as practically useful as possible. We are still a little uncomfortable with this approach as it feels anecdotal for the scholarly literature. The paper was intended very much as a review article to outline the relevant approaches in this field, with practical examples and insights from published literature, not a primary piece of research. We therefore haven't tested the approach proposed empirically. We have however added a table 2 which gives an example of how we applied this process on a species in our institutions. We hope this makes it clearer how one might go about the process we have proposed in deriving welfare assessment tools. in light of our concern with this inclusion we have added a sentence in the discussion to caveat that this process has not been empirically tested. This sentence says 'The thinking and steps outlined in Fig 2 and Table 2 have not been empirically tested and there remains a future need to evaluate the effectiveness of this process across a few different species and tool types. '